# Antioxidant Agriculture for Stress-Resilient Crop Production: Field Practice

**DOI:** 10.3390/antiox13020164

**Published:** 2024-01-27

**Authors:** Yao Sun, Xianzhi Xie, Chang-Jie Jiang

**Affiliations:** Shandong Academy of Agricultural Sciences, Jinan 250100, China; sunyao0525@gmail.com (Y.S.); xzhxie2010@163.com (X.X.)

**Keywords:** reactive oxygen species (ROS), oxidative stress, antioxidant defense, stress tolerance, crop productivity, yield

## Abstract

Oxidative stress, resulting from the excessive production of reactive oxygen species, is a common and major cause of cellular damage in plants exposed to various abiotic stresses. To address this challenge, we introduce the concept of antioxidant agriculture as a comprehensive strategy to improve stress tolerance and thus crop productivity by minimizing oxidative stress levels in the field environment. This strategy encompasses a diverse range of approaches, including genetic engineering, the exogenous application of antioxidant agents, microbial inoculation, and agronomic practices, to reinforce the plant’s intrinsic antioxidant defense system and mitigate oxidative stress. We present recent successful studies of antioxidant measures that have been validated in field conditions, along with our perspective on achieving antioxidant agriculture.

## 1. Introduction

Plants constantly face various abiotic stresses in the field environment due to their sessile lifestyle such as drought, salinity, temperature extremes, and nutrient deficiency [1]. Additionally, climate change is intensifying the frequency and magnitude of the stresses [1,2,3]. These stresses impact crop plant growth, resulting in substantial annual yield losses and posing a significant challenge to food security [1]. A global survey indicated an annual loss of crop yield ranging from 51% to 82% worldwide due to abiotic stresses [4]. Simultaneously, the global human population is continuously increasing and is projected to reach 9–10 billion by 2050 [1,2]. Therefore, urgent measures are required to comprehensively mitigate the adverse effects of these stresses, ensuring sustainable agricultural development to meet the growing food demand.

One common response of plants to various abiotic stresses is the production of reactive oxygen species (ROS) [5]. These ROS serve a dual function: acting as signaling molecules in stress responses at low concentrations, and causing cellular damage at excessive levels. It has been demonstrated that overaccumulation of ROS leads to oxidative stress, contributing to molecular and cellular damage in plants under various stresses [5]. To counteract oxidative stress, plants have evolved a complex antioxidant system comprising enzymatic and nonenzymatic antioxidants that scavenge ROS and protect the cellular components from oxidation [5,6]. Numerous studies have explored diverse antioxidant strategies that can mitigate oxidative stress, enhancing stress tolerance and plant productivity [5,7,8]. However, few of these antioxidant measures have been translated into agricultural field application, emphasizing the need to bridge the gap between laboratory research and field implementation.

In this respect, we introduce the concept of antioxidant agriculture, a comprehensive strategy to mitigate oxidative stress and improve crop growth and productivity under abiotic stresses in the field environment. We present recent field-validated studies on antioxidant crop management and discuss strategies for bridging the gap between laboratory research and the field application of antioxidant agriculture.

## 2. Oxidative stress in plants

Reactive oxygen species (ROS) are chemically reactive molecules that contain oxygen atoms with one unpaired electron. ROS include superoxide anions (O_2_^•−^), hydrogen peroxide (H_2_O_2_), hydroxyl radical (^•^OH), and singlet oxygen (^1^O_2_). They are produced constantly as byproducts of normal cellular metabolism in almost every subcellular compartment, but primarily in the chloroplasts, mitochondria, and peroxisomes in plants [5,9]. The NADPH oxidase (respiratory burst oxidase homolog: RBOH) is a key enzyme in ROS production in plants [10].

ROS act as signaling molecules to regulate various cellular processes, such as gene expression, hormone synthesis, cell division, and cell death, important for plant growth, development, and stress responses. However, ROS can also cause damage to plant cells if they accumulate in excess. They can damage the structure and function of lipids, nucleic acids, and proteins, leading to oxidative stress. Therefore, plants must maintain a delicate balance between ROS production and removal in order to avoid oxidative stress [9].

Plants have evolved a complex antioxidant system to scavenge excess ROS and protect cellular components from oxidative damage. The antioxidant system includes both enzymatic and nonenzymatic antioxidants. Enzymatic antioxidants include superoxide dismutase (SOD), catalase (CAT), ascorbate peroxidase (APX), nonspecific peroxidase (POX), monodehydroascorbate reductase (MDHAR), monodehydroascorbate reductase (DHAR), glutathione *S*-transferase (GST), glutathione peroxidase (GPX), alternative oxidase (AOX), and peroxiredoxin (Prx) [9,11]. Nonenzymatic antioxidants include ascorbic acid (AsA), glutathione (GSH), carotenoids, tocopherols, and flavonoids [9]. Each of these antioxidants has a unique mechanism of action and can target different types of ROS.

Oxidative stress occurs when ROS production exceeds the capacity of a cell’s antioxidant system to neutralize them, and it can impair the photosynthesis, respiration, and nutrient uptake of plants and reduce their growth and yield (Figure 1).

## 3. The Impact of Oxidative Stress on Plants

Under normal plant growth conditions, ROS production and removal are in balance, and the levels maintained are low. However, when plants are exposed to continuous abiotic stresses, ROS production can drastically increase, disturbing the normal balance [9]. Studies have shown that almost all kinds of abiotic stresses cause ROS to overaccumulate, which damages cells and inhibits plants growth. For example, a significant increase in ROS levels has been observed in major crop plants when exposed to various abiotic stresses including drought, salinity, alkali conditions, cold, heat, flooding, heavy metals, nutrient imbalance, UV, and ozone exposure [5,9,12]. This increase in ROS levels is a major route to plant damage and growth inhibition.

Plants activate a series of antioxidant reactions in response to stress exposure to reduce excessive ROS accumulation. Studies have shown that the enzyme activity of SOD, CAT, APX, and POX rapidly and significantly increases in response to various stresses [11,12]. Additionally, the nonenzymatic antioxidant system is also activated by increasing the levels of various endogenous antioxidants, such as AsA, GSH, carotenoids, tocopherols, and flavonoids [9]. However, the capacity of the antioxidant defense system is often overwhelmed by the increased ROS production activity, resulting in oxidative stress (Figure 1). On the other hand, many studies have shown that stress damage to plants can be mitigated by taking various artificial antioxidant measures to increase antioxidant activity [8].

These results demonstrate that the harmful effects of most abiotic stresses are associated with ROS overaccumulation due to the collapse of the homeostatic capacity at some point. Therefore, minimizing excess ROS accumulation is key to mitigating stress damage to plant cells and enhancing crop productivity under stressful conditions.

## 4. Antioxidant Agriculture

Based on the above, we introduce the concept of antioxidant agriculture. This concept refers to practical field applications that minimize excessive ROS accumulation caused by stresses, thereby enhancing crop productivity in agricultural fields (Figure 1). It may encompass a range of strategies that enhance the antioxidant capacity of plants, such as genetic engineering, chemical application, and agronomic practices.

To date, numerous studies have produced genetically modified crops with enhanced antioxidant activity through molecular manipulation of antioxidant-related enzymes and transcription factors [7,8,13]. These genetically modified crops have demonstrated improved tolerance to various abiotic stresses. Furthermore, hundreds of chemicals, including natural and synthetic antioxidant compounds, phytohormones and their functional analogs, and plant biostimulants, have been shown to activate plant antioxidant machinery and/or reduce ROS accumulation, thereby enhancing plant growth and productivity [14,15,16]. Additionally, methods employing microorganisms and stress-hardening procedures have also been reported to be effective in improving tolerance to oxidative stress [17].

However, most of this research remains confined to the controlled environment of the laboratory or greenhouse setting, with only a limited number of studies having been translated into actual field trials. Consequently, the current situation is that many research findings that enhance the antioxidant capacity of plants have not yet been successfully implemented in crop production under stress conditions in field settings.

In this section, we present recent successful examples validated through open field trials that demonstrate enhanced antioxidant capacity and improved crop stress tolerance (Table 1). These examples are selected based on a literature search conducted using the Google Scholar search engine and do not necessarily cover all relevant studies.

### 4.1. Genetic Engineering

Oxidative stress can be mitigated by reinforcing the ROS scavenging system through genetic engineering of the relevant functional genes. These genes encompass those involved in both enzymatic and nonenzymatic ROS scavenging, as well as their transcriptional regulation.

SOD is a crucial enzyme in the antioxidant mechanism, catalyzing the dismutation of O_2_^•−^ into H_2_O_2_ and molecular oxygen (O_2_) [8]. Overexpression of a cDNA encoding manganese SOD (Mn-SOD) from *Nicotiana plumbaginifolia* in alfalfa (*Medicago sativa* L., cv R3), with the resulting protein targeted to either chloroplasts or mitochondria, has been shown to enhance resistance to acifluorfen-induced oxidative stress, as evidenced by a significant reduction in electrolyte leakage in leaves during freezing and water-deficit stresses [18,19]. A three-year field trial demonstrated that the yield of these transgenic plants was approximately double that of the nontransgenic control (R3) in the first and second years, and three to five times higher in the third year following two winters [19]. This study underscores the successful translation of an antioxidant-related gene, initially discovered and tested in model plant species, into crop species.

GSH, a low molecular weight tripeptide (Glu–Cys–Gly), is one of the most crucial and potent antioxidant molecules in the nonenzymatic antioxidant mechanism. GSH synthesis is mainly catalyzed by γ-glutamylcysteine synthetase (γ-ECS) and glutathione synthetase (GS) [8]. In rice (*Oryza sativa* L.), overexpression of a cDNA encoding γ-ECS from *Brassica juncea* under the stress-inducible *Rab21* promoter has been demonstrated to mitigate paraquat-induced oxidative cellular damage and enhance tolerance to high salinity by maintaining the cellular GSH/oxidized glutathione (GSSG) ratio [20]. These transgenic rice plants displayed a moderate increase in biomass and grain yield under paddy field conditions when compared to their wild type plants. Furthermore, transgenic rice plants overexpressing the *OsGS* gene under the constitutive *OsCc1* promoter also displayed a higher GSH/GSSG ratio and an improved grain yield and total biomass than their wild type plants in a two-year paddy field trial [21].

The *Arabidopsis DROUGHT TOLERANT PROTEIN 6* (*AT-DTP6*) gene was identified through activation tag screening for tolerance to soil drought stress [22]. Overexpression of this gene reduced DAB staining and H_2_O_2_ levels in both *Arabidopsis* and maize (*Zea mays* L.) plants exposed to paraquat, implying an antioxidant role for *AT-DTP6* [23]. The maize transgenic plants overexpressing *AT-DTP6* were field-tested for three years and showed an average 4% increase in grain yield compared to the control under severe drought conditions [23]. These results suggest that *AT-DTP6* is a promising target for engineering stress tolerance in crops.

The CRISPR/Cas9 system has emerged as a powerful and precise genome editing tool, holding immense potential for developing stress-tolerant crops and enhancing agricultural productivity [24]. CRISPR/Cas9-mediated null mutagenesis of the rice *PARAQUAT TOLERANCE 3* (*OsPQT3*) led to increased expression of antioxidant enzyme genes *OsGPX1*, *OsAPX1,* and *OsSOD1*, thereby enhancing rice resistance to oxidative and salt stresses [25]. *OsPQT3* encodes an E3 ubiquitin ligase that negatively regulates the response to oxidative stress. Remarkably, under salt stress conditions, the mutant plants exhibited significantly enhanced grain yield compared with the wild type, in both greenhouse and field settings [25].

Recently, Zhang, et al. [26] identified a gene in the sorghum genome, *Alkaline Tolerance 1 (AT1*), that determines saline–alkaline sensitivity. The *AT1* gene encodes an atypical G protein g subunit that negatively regulates the phosphorylation of aquaporin PIP2;1, leading to increased ROS content in plants under alkaline stress. CRISPR/Cas9-mediated knockout of this gene greatly reduced the alkali-induced H_2_O_2_ accumulation. In a field test conducted in highly sodic soil, genetically engineered sorghum, millet, rice, and maize plants with knockouts or natural nonfunctional alleles of *AT1* homologs showed significantly enhanced tolerance to alkaline stress, resulting in a 20–28% increase in grain yield when compared with wild type plants. These results demonstrate that genetic engineering of *AT1* homologs can significantly contribute to increasing crop production in sodic lands.

These findings collectively suggest that enhancing crop tolerance to diverse stresses can be achieved through molecular modification of a gene involved in the antioxidant mechanism. The targeted gene can be either an endogenous gene or a foreign gene. However, transgenic crops continue to face regulatory and societal challenges in many countries due to potential risks to human health and the environment. A promising solution to this issue is transgene-free genome editing technology, which is gaining increasing interest as it does not leave any trace of foreign DNA or editing components in the plant genome [24,27]. Similar to mutagenized crops created using chemical or physical agents to induce random mutations, genome-edited plants without a transgene can be readily introduced into the field without restrictions.

Additionally, substantial variations in antioxidant defense capacity have been observed across plant species and genotypes [5]. This finding suggests that conventional breeding programs can play a significant role in developing varieties with enhanced antioxidant properties.

### 4.2. Chemical Application

A diverse array of chemicals, including natural and synthetic antioxidants, phytohormones, and plant biostimulants, has proven effective in mitigating oxidative stress during crop cultivation in challenging field conditions. Among practical application methods, foliar spraying and seed treatment are the most commonly used [16]. Compared to genetic engineering, a chemical approach, particularly the use of antioxidants and biostimulants, presents a potentially faster route to antioxidant agriculture as it bypasses both the stringent regulatory approval processes and the societal controversies associated with genetically modified crops.

#### 4.2.1. Antioxidants

Antioxidants are molecules that can neutralize ROS, thereby preventing potential damage to cellular components [16]. Exogenous application of antioxidants, such as those found in plant-sourced extracts or synthetic formulations, can provide an additional line of antioxidant defense, complementing the plant’s own antioxidant defense system.

AsA (vitamin C) and GSH are among the most abundant antioxidants in plants. Foliar application of AsA to wheat (*Triticum aestivum* L.) plants at stem elongation and booting stages has been shown to significantly increase the activities of CAT and POX enzymes in drought-stressed fields. This increase led to an increase in leaf relative water content (RWC), chlorophyll content, and grain yield [28]. Furthermore, foliar spraying with AsA [29] and GSH [30] on common bean (*Phaseolus vulgaris* L.) seedlings significantly improved vegetative plant growth and green pod yield in drought-stressed fields. Mechanistic analyses revealed that these improvements are associated with increased activity of antioxidant enzymes and endogenous contents of antioxidants like GSH, AsA, proline, and total soluble sugars.

Tocopherols are a class of lipid-soluble antioxidants, belonging to the vitamin E family [31]. Sadiq et al. [32] reported that foliar application of α-tocopherol on mung beans (*Vigna radiata* L.) at the vegetative stage significantly increased the activity of key antioxidant enzymes (SOD, POD, and CAT), leading to a reduction in toxic levels of H_2_O_2_ and malondialdehyde (MDA, a biomarker of lipid peroxidation caused by ROS) in drought-stressed fields. These positive changes resulted in increased plant biomass and seed yield. Similarly, Ali et al. [33] found that foliar application of α-tocopherol on wheat plants at the heading stage significantly enhanced their drought tolerance in the fields. This was evident in the improved enzymatic and nonenzymatic antioxidant defense mechanisms, plant growth, and grain yield [33]. Interestingly, α-tocopherol application also improved the seed nutritional quality by increasing the content of seed phenolics, flavonoids, and tocopherols [33].

Proline, a proteinogenic amino acid, is a key organic osmolyte. It has also been shown to scavenge free radicals generated in plants under various stress conditions [34]. Foliar spraying with proline on sugar beet seedlings increased SOD and CAT activity, leading to a decrease in ROS and MDA levels. This antioxidant effect significantly mitigated the adverse effects of drought stress, as evidenced by increased photosynthetic pigments, RWC, membrane stability, and root and sugar yield in a two-year field trial [35,36]. Proline alone or combined with silicon (Si) showed a similar improvement in drought tolerance [36]. Similarly, in barley (*Hordeum vulgare* L.), foliar application of proline and salicylic acid (SA) enhanced the enzyme activity of CAT, POX, and polyphenol oxidase (PPO), and decreased O_2_^•−^ and H_2_O_2_ accumulation, leading to increased chlorophyll content, RWC, plant dry mass, and grain yield in drought-stressed fields [37].

γ-aminobutyric acid (GABA) is a nonprotein amino acid, functioning as both a metabolite and a signal in response to biotic and abiotic stresses [38]. Foliar application of GABA has been shown to improve the tolerance of black cumin (*Nigella sativa* L.) [39] and common bean (*Phaseolus vulgaris* L.) [40] to drought stress by increasing the activity of the antioxidant defense system, reducing ROS accumulation, and ultimately leading to increased plant growth and seed yields.

Caffeic acid (CA), a phenolic compound, has been shown to scavenge ROS [41]. Foliar application of CA to wheat plants significantly reduced salt stress-induced accumulation of MDA and H_2_O_2_ as well as Na^+^ uptake [42]. This led to improvements in nutrient uptake, plant biomass, and grain yield.

These results indicate that the exogenous application of antioxidants is a promising strategy for reducing oxidative stress and improving plant growth in stressful field conditions. However, while commercial production systems for the above-mentioned antioxidants are well established and the products are widely used as supplements for human health, the concentrations employed in the field experiments were relatively high—reaching millimolar levels in many cases. This fact raises concerns about potential cost limitations for large-scale agricultural applications. Therefore, further field analyses are likely to be required to assess the feasibility and applicability of these antioxidants in practical agricultural settings.

#### 4.2.2. Phytohormones

Phytohormones have been extensively studied and widely used in agriculture. Their exogenous application has shown promise in mitigating stress-induced damage on plants.

Abscisic acid (ABA) plays a key role in plant response to various abiotic stresses. Wei et al. [43] pretreated rice seedlings with ABA for 24 h through root-drenching before transplanting them in saline–alkaline paddy fields. A three-year field trial showed that the priming treatment with ABA significantly improved tolerance to saline–alkaline stress, leading to increased seedling survival rate, plant growth, and final grain yield by 8–55% compared with the control treatment. Mechanistically, the exogenous ABA enhanced the activity of antioxidant enzymes SOD, CAT, APX, and POD, thereby reducing ROS accumulation [44].

In drought-stressed fields, exogenous application of ABA and cytokinin (6-benzyladenine, BA), either individually or in combination, to wheat plants after the anthesis stage increased glycine betaine and proline contents, as well as the enzyme activities of CAT and POX. This led to a reduction in the accumulation of H_2_O_2_ and MDA [45].

Additionally, foliar application of jasmonic acid to sugar beet plants significantly increased antioxidant enzyme activities, root yield, and white sugar content (primarily composed of sucrose) [46].

Furthermore, Ghasemi et al. [47] recently reported that foliar application of 24-epibrassinolide (EBL), spermine, and Si to maize plants enhanced the enzyme activities of SOD, CAT, and POX, as well as the levels of phenolic antioxidants, proline, and glycine betaine. These antioxidant effects ultimately led to an increase in plant growth and grain yield. Notably, the application of this triad exhibited an even greater improvement in plant growth and grain yield compared to individual applications.

These findings highlight the potential of phytohormones as effective tools for mitigating oxidative stress and improving plant growth under challenging environmental conditions. In particular, the exogenous application of ABA has shown significant promise in improving yield performance under various stress conditions. Notably, the recent discoveries of functional analogs of ABA and inhibitors of ABA metabolism [48,49] will further promote the practical application of compounds that activate the ABA pathway to enhance crop productivity in stressed agricultural fields.

#### 4.2.3. Plant Biostimulants

A plant biostimulant (also known as biofertilizer or plant enhancer) is a substance or microorganism that improves plant growth and development, quality, and stress tolerance by stimulating natural processes [15]. Some plant biostimulants have been shown to enhance antioxidant activity in various crop plants [14,50].

Seaweeds have been used as a soil amendment since ancient times to improve crop productivity. Seaweed extracts (SWEs) are rich in diverse bioactive compounds, such as minerals, proteins and peptides, polysaccharides, phenolic compounds, phytohormones, and antimicrobials. SWEs are among the most widely used biostimulants, capable of improving plant performance and productivity under both normal and various stress conditions [51]. While the efficacy of many SWEs has been demonstrated, particularly at laboratory levels, the mechanisms underlying their action remain poorly understood. In maize plants, foliar application of SWE derived from the brown algae *Ascophyllum nodosum* mitigated drought stress, improved plant growth, and enhanced source-to-sink translocation of carbohydrates. This process resulted in a 3.08 t/ha increase in stalk yield and a 3.4 kg/t increase in sugar yield. Physiological analysis revealed that SWE application significantly enhanced antioxidant enzyme activity, reduced MDA levels, and increased metabolic activity [52].

Humic acid, a key component of humic substances found in soil, sediment, and aquatic environments, has been shown to improve plant performance and productivity through modulating their antioxidant systems [53]. Kaya et al. [54] investigated the field-based effectiveness of leonardite (containing up to 90% humic acid; Ankara, Turkey) and pure humic acid as soil amendments for enhancing tolerance to drought and phosphorus deficiency in maize plants. Both drought and phosphorus deficiency were found to reduce plant growth, yield, and chlorophyll content. However, leonardite and humic acid, each applied in combination with sulfur, significantly improved plant growth and yield under both stress conditions. Notably, the largest improvement was observed under combined stress of drought and phosphorus deficiency. This improvement was attributed to enhanced antioxidant activity and increased leaf phosphorus and leaf RWC.

β-sitosterol, a plant sterol with a chemical structure similar to cholesterol, has been shown to improve plant performance and productivity under various biotic and abiotic stresses by regulating both the ROS balance and the source-to-sink relationship [55]. Foliar application of β-sitosterol on wheat plants enhanced antioxidant activity in drought-stressed fields, as evidenced by the increased activity of antioxidant enzymes (SOD, CAT, POD, and APX), as well as elevated levels of AsA, tocopherol, proline, and carotene. Consequently, H_2_O_2_ and MDA levels were reduced. This increased antioxidant potential ultimately led to increased plant biomass and grain yield [56].

The root of the licorice plant (*Glycyrrhiza glabra* or *Glycyrrhiza uralensis*) has long been used in traditional Eastern and Western medicine. Licorice root extract (LRE) contains many beneficial compounds, such as glycyrrhizin, glabridin, licochalcone A, licoricidin, and licorisoflavan A, that have antioxidant, anti-inflammatory, antibacterial, and antiviral properties [57]. In common bean plants, seed soaking and/or foliar spray with LRE significantly improved plant growth and yield under salt stress [58]. These improvements were accompanied by increased levels of photosynthetic pigments, free proline, total soluble carbohydrates, K+/Na+ ratio, and RWC, while reducing O_2_^•−^, H_2_O_2_, MDA content, and electrolyte leakage. Combining seed soaking and foliar spray yielded the best results.

Silicon (Si) is considered a nonessential element for plant growth and development. However, exogenous application of Si has long been known to provide many benefits to plants, especially under various biotic and abiotic stresses [59]. Thus, Si is often considered as a plant biostimulant. As described in above sections (Section 4.2.1 and Section 4.2.2), foliar application of Si to sugar beet [36] and maize plants [47] has been shown to enhance antioxidant potential and plant growth.

In summary, plant biostimulants are gaining increasing attention from both the scientific and agricultural communities due to their potential to improve plant growth, development, quality, and stress tolerance, thereby contributing to sustainable agriculture and plant science. Nonetheless, further research is needed to optimize the dosage, timing, and application methods for effective field implementation.

### 4.3. Microbial Application

Plant-beneficial microbial agents that consist of one or more microorganism, primarily bacteria, fungi, and/or algae, can promote plant growth and health in various ways [17]. Seed inoculation with a mung bean rhizosphere-associated *Pseudomonas aeruginosa* GGRJ21 strain has been shown to elevate antioxidant enzyme activity (SOD, CAT, and POX) and proline content, and increase the expression of drought stress-responsive genes, RWC, root and shoot lengths, and biomass in drought-stressed fields [60]. Furthermore, Iqbal et al. [61] reported that seed inoculation of maize with auxin-producing rhizobacteria (strain MA4 and MA11) improved phosphorous uptake, plant growth, and grain yield (31%) under salt stress conditions. However, contrarily, the proline content and APX and SOD activity in leaves declined. This reduction in antioxidant activity may be attributed to the fact that these rhizobacterial strains relieved the plants from the deleterious effects of salinity.

Microbial agents offer a viable alternative or supplemental strategy to support plants facing abiotic stresses. However, their effectiveness varies significantly depending on factors like plant species, environmental conditions, and soil characteristics. This necessitates careful optimization of microbial agent use to maximize their impact on crop stress tolerance and productivity. Furthermore, potential negative impacts on the native microbial community and ecological balance require thorough investigation before widespread application.

### 4.4. Agronomic Practice

Wang et al. [62] conducted a three-year field trial to explore the impact of tillage on rapeseed (*Brassica napus* L.) seedling conditions during an overwintering period and on seed yield. This study revealed a remarkable finding: compared to shallow tillage (ST), moderate deep tillage (MT) significantly enhanced the activity of key antioxidant enzymes (SOD, CAT, POD, and APX), while reducing O_2_^•−^, H_2_O_2,_ and MDA content in the roots and leaves during the overwintering stage. Additionally, MT improved rapeseed seedling conditions during the overwintering period and ultimately increased the seed yield by 23.5% compared to ST. This promising finding suggests that adapting agronomic practices can be a valuable tool for enhancing antioxidant activity and stress resilience, ultimately increasing crop productivity.

**Table 1 antioxidants-13-00164-t001:** Field-validated antioxidant measures for improved crop stress tolerance.

Strategies	Crop Species	Stress Conditions	Antioxidant Agents	Field Location Years of Trial	Oxidative Stress Status and Plant Performance	References
Genetic engineering	Alfalfa	Drought	Transgenic OX of *Mn-SOD*	Elora, Ontario, Canada; 3 years (1992–1994)	↑ Resistance to acifluorfen-induced oxidative stress and freezing tolerance ↑ Plant biomass	[18,19]
	Rice	High salinity	Transgenic OX of—*BrECS*	Daegu, South Korea	↑ Resistance to paraquat-induced oxidative stress and GSH/GSSG ratio ↑ Plant biomass and grain yield	[20]
	Rice	High salinity	Transgenic OX of *OsGS*	Gunwi, South Korea; 2 years (2014–2015)	↑ GSH/GSSG ratio	[21]
	Maize	Drought	Transgenic OX of *AT-DTP6*	Woodland, California; 3 years	↓ DAB staining and H_2_O_2_ content ↑ Grain yield	[23]
	Rice	Oxidative and salt stresses	Knockout of *OsPQT3*	Greenhouse	↑ Tolerance to oxidative and salt stresses	[25]
	Sorghum, millet, maize, and rice	Alkalinity	Knockout of *AT1* homologs	Ningxia and Jilin, China; Sorghum, millet and maize, 1 year (2021); rice, 2 years (2021–2022)	↓ DAB staining and H_2_O_2_ content ↑ Biomass and grain yield	[26]
Antioxidants	Wheat	Drought	Ascorbic acid, FA at stem elongation and booting stages	Kafr Elsheikh, Egypt; 2 years (2012–2014)	↑ CAT and POX activities, RWC, and chlorophyll content ↑ Grain yield	[28]
	Common bean	Drought	Ascorbic acid, FA	Nubaria, Egypt; 1 year (2017)	↑ Vegetable plant growth and green pod yield	[29]
	Common bean	Drought	Glutathione, FA	Fayoum Governorate, Egypt; 2 years (2017–2018)	↑ SOD, CAT, APX, and GSH-Px activities ↑ Vegetable plant growth and green pod yield	[30]
	Mung bean	Drought	α-tocopherol, FA	Jhang, Punjab, Pakistan	↑ SOD, POD, and CAT activities ↓ H_2_O_2_ and MDA levels	[32]
	Wheat	Drought	α-tocopherol, FA at heading stage	Faisalabad, Pakistan	↑Activity enzymatic and nonenzymatic antioxidant defense mechanisms ↑ Plant growth and grain yield	[33]
	Sugar beet	Drought	Proline, FA at seeding stage	Gharbia Governorate, Saudi Arabia; 2 years (2018–2020)	↑SOD and CAT activities ↓ ROS and MDA levels	[36]
	Sugar beet	Drought	Proline, FA at seeding stage	Chaha-rmahal-Bakhtiari province, Iran; 2 years (2014–2015)	↑ SOD and CAT activities ↓ROS and MDA levels	[35]
	Barley	Drought	Proline, FA	Kafr el-Sheikh Governorate, Egypt; 2 years (2017–2018)	↑ CAT, POX, and PPO activities, chlorophyll content, RWC, O_2_^•−^, and H_2_O_2_ levels ↓ Plant dry mass and grain yield	[37]
	Black cumin	Drought	γ-aminobutyric acid, FA	Naqadeh-Urmia, West Azerbaijan, Iran; 2 years	↓ ROS and MDA levels ↑ Plant growth and seed yields	[39]
	Snap bean	Drought	γ-aminobutyric acid, FA	Wadi El-Natroun, Beheira Governorate, Egypt; 2 years (2018–2019)	↓ ROS and MDA levels ↑ Plant growth and seed yields	[40]
Plant hormones	Barley	Drought	Salicylic acid, FA	Kafr el-Sheikh Governorate, Egypt; 2 years (2017–2018)	↑ Activity of CAT, POX, and PPO, chlorophyll content, RWC, O_2_^•−^, and H_2_O_2_ levels ↑ Plant dry mass and grain yield	[37]
	Rice	High salinity	ABA, priming of seedlings for 24 h before transplanting	Da’an, Jilin, China; 3 years (2012–2014)	↓ ROS and MDA levels ↑ Seedling survival rate, plant growth, and final grain yield	[43,44]
	Wheat	Drought	ABA, 6-BA (cytokinin)	Tehran, Iran; 2 years (2009–2010)	↑ Glycine betaine and proline contents, and CAT and POX activities ↓ H_2_O_2_ and MDA levels	[45]
	Sugar beet	Drought	JA, FA	Bakhtiari province, Iran; 1 years (2015)	↑ Activity of antioxidant enzyme activities, root yield, and white sugar content	[46].
	Maize	Drought	24-epibrassinolide, FA	Moghan, Iran; 2 years (2020–2021)	↑ SOD, CAT, and POX activities, chlorophyll content, and RWC ↓ H_2_O_2_ and MDA levels ↑ Plant growth	[47]
Plant biostimulants	Sugarcane	Drought	Seaweed extracts, FA	Bunge mill, Dourados, Brazil, 2018 São Martinho mill, Pradópolis, Brazil, 2019 São Martinho mill, Motuca, Brazil, 2020	↑ SOD, CAT, and POX activities ↑ Metabolic activity ↓MDA and H_2_O_2_ levels	[52]
	Maize	Drought Phosphorus deficiency	Sulfur-enriched leonardite	Sanliurfa, Turkey; 1 year (2011)	↑ Antioxidant activity, chlorophyll content, F_v_/F_m_, and RWC ↓ Electrolyte leakage and leaf H_2_O_2_ levels ↑ Plant biomass and grain yield,	[54]
	Wheat	Drought	β-sitosterol, FA	Nubaria region, Egypt; 2 years (2016–2017)	↑ SOD, CAT, and POX activities ↑ Metabolic activity ↓MDA and H_2_O_2_ levels	[56]
	Common bean	High salinity	Licorice root extract, seed soaking/FA	El-Noubaria, Egypt; 3 years (2015–2017)	↓ O_2_^•−^, H_2_O_2_, and MDA levels ↑ Photosynthetic pigments, proline, total soluble carbohydrates, K+/Na+ ratio, and RWC	[58]
	Sugar beet	Drought	**Si**, FA at seeding stage	Gharbia Governorate, Saudi Arabia; 2 years (2018–2020)	↑ SOD and CAT activities ↓ ROS and MDA levels	[36]
	Maize	Drought	Si, FA	Moghan, Iran; 2 years (2020–2021)	↑ SOD, CAT, and POX activities, chlorophyll content, and RWC ↓ H_2_O_2_ and MDA levels ↑ Plant growth	[47]
Microbial agents	Mung bean	Drought	*Pseudomonas aeruginosa* GGRJ21	Jorhat district, Assam, India; 3 years (2011–2013)	↑ SOD, CAT, and POX activities, proline content, expression of drought stress-responsive genes, and RWC ↑ Root and shoot lengths, and biomass	[60]
	Maize	High salinity	Rhizobacterial strains (MA4 and MA11)		↑ Phosphorous uptake ↑ Plant growth and grain yield ↓ Proline content, and APX and SOD activities	[61]
Agronomic practice	Rapeseed	Low temperature	Moderate deep tillage	Wuhan, Hubei Province, China; 3 years	↑ SOD, CAT, POD, and APX activities ↓O_2_^•−^, H_2_O_2_, and MDA levels ↑ Rapeseed seedling conditions during overwintering period and yield	[62]

OE: overexpression; FA: foliar application; *F_v_/F_m_*: maximum fluorescence yield; upward (↑) and downward (↓) arrows indicate increases and decreases relative to the control, respectively.

## 5. Perspectives

The generation of ROS in plant cells is a common response to diverse abiotic stresses. These ROS act as signaling molecules or damaging agents, depending on their concentration. Excessive ROS accumulation has been linked to the harmful effects of various stresses. Therefore, antioxidant measures are expected to confer cross-tolerance to multiple stresses. This makes the development and implementation of antioxidant agriculture highly beneficial for achieving stress-resilient and sustainable crop production, as crop plants routinely face multiple concurrent stresses in the field environment. We suggest that antioxidant agriculture is a promising and viable solution to improve stress-resilient crop production in the face of global climate change and environmental degradation.

While a wealth of research on oxidative stress exists at the laboratory level, translating these findings into effective field applications remains a significant challenge. Antioxidant agriculture requires balancing the tradeoff between stress tolerance and plant growth. Therefore, it is vital to bridge the gap between laboratory research and field application, evaluating the feasibility and practicality of the antioxidant measures developed in the laboratories and implementing them effectively in real-world crop production in field settings. To achieve this aim, close cooperation among relevant sectors, including research institutes, industry, and farmers, is essential. Additionally, recognizing that field trials often generate fewer publications than other research types is crucial for ensuring appropriate recognition and evaluation of researchers engaged in this vital work.

## Figures and Tables

**Figure 1 antioxidants-13-00164-f001:**
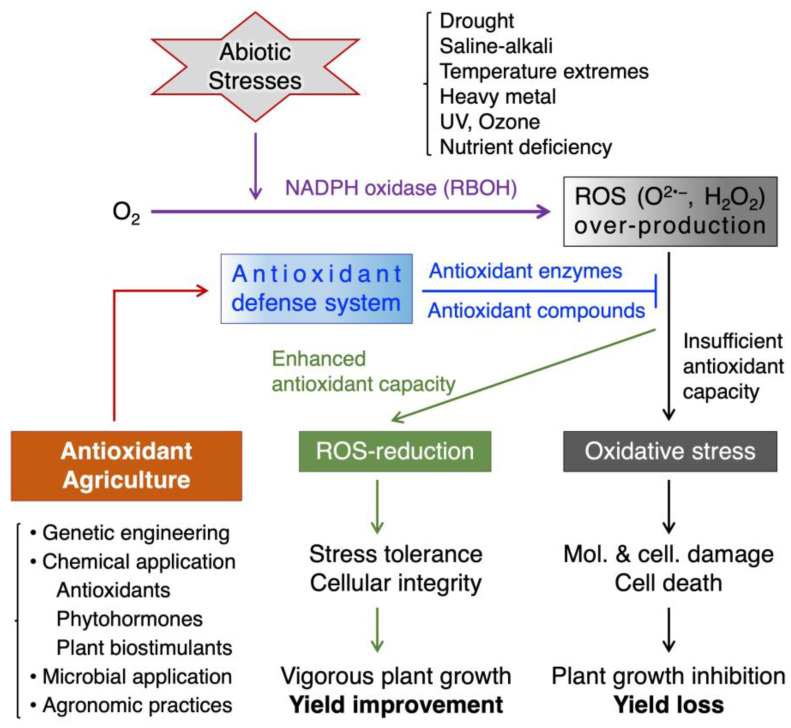
Abiotic stresses, ROS, and antioxidant agriculture in crop plants. Abiotic stresses trigger the production of ROS in plants (purple arrows). ROS act as signaling molecules to activate stress responses at low levels (green arrows), but they also cause oxidative damage to biomolecules and cellular structures at high levels (black arrows). Plants have an intrinsic antioxidant defense system that can scavenge excess ROS and prevent oxidative stress (blue lines). However, when ROS production exceeds the antioxidant capacity, oxidative stress occurs and impairs plant growth and productivity (black arrows). Antioxidant agriculture is a strategy that aims to enhance the antioxidant defense system of plants through strategies including genetic engineering of antioxidant enzymes, exogenous application of antioxidant agents, phytohormones, plant biostimulants, beneficial microbial inoculation, and agronomic practices (red arrow). This can help reduce oxidative stress (blue lines) and improve plant stress tolerance and productivity in agricultural fields (green arrows). Arrows and lines with a bar at the end indicate positive and negative regulation, respectively.

## Data Availability

Data sharing is not applicable to this article as no new data were created or analyzed in this study.

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
