# Peer review of "Antioxidant Agriculture for Stress-Resilient Crop Production: Field Practice"

_antioxidants, 2024, doi:10.3390/antiox13020164_

Round 1
Reviewer 1 Report
Comments and Suggestions for Authors
Dear Authors,
Reviewer comments antioxidants-2819707
The manuscript entitled „Antioxidant agriculture for stress-resilient crop production: The field practice“ represents a useful review manuscript on the concept of so-called antioxidant agriculture as a whole of means and practices aimed to alleviate the adverse effects of oxidative stress on crop plants. The review manuscript provides data on the utilization of genetic engineering, application of several bioactive compounds including antioxidants, phytohormones, plant biostimulants, and agronomic practice to improve crop tolerance to oxidative stress. A comprehensive overview of the studies using various antioxidant agents to alleviate adverse impacts of environmental stresses, especially drought, high salinity, and low temperature in a wide range of crops including alfalfa, barley, wheat, rice, maize, common bean, sugar beet, and rapeseeds is provided in Table 1.
I think that the manuscript summarises very useful data importnat for further research and thus it deserves to be published in Antioxidants.
However, I have a few important comments on the present version of the manuscript which are provided below:
The concept of antioxidant argiculture discussed in the text should be described more clearly. What are the inputs and the outputs of antioxidant agriculture? In addition to the text and Table 1 providing just an overview on published literature on this topic, I would strongly recommend the authors to add a figure (Figure 1) providing a scheme summarising the concept of antioxidant agriculture considering its inputs (genetic engineering, treatments with antioxidant agents and bioactive compounds, agricultural management practices) and outputs (improved crop stress tolerance).
Formal comments on the text related to terminology, English language and style:
Lines 154-155: Use „gamma“ letter instead of „g“ for „γ-glutamylcysteine synthetase (γ-GCS)“, not „g-ECS“ – please, correct the spelling of the abbreviation commonly used for the enzyme!
Line 192: Add either the word „fact“ or „finding“ in the statement: „This finding suggests that conventional breeding programs can play a significant role in….“
Line 258: Add the word „finding“ following „This“ in the statement: „This finding raises concerns about potential cost limitations…“
Line 278: Briefly define „white sugar content“, i.e., „sucrose“???
Line 297: The sceintific name for „seaweeds“ has to be given.
Line 306: Add the word „process“ in the statement „This proces resulted in a 3.08 t/ha increase….“
Line 315: The term „leonardite“ (containing up to 90% humic acid) probably refers to some commercial means used in the study. Therefore, an appropriate reference including the manufacturer – company name – has to be given for leonardite.
Line 389: Add the word „aim“ following „this“ in the statement „To acheive this aim,….“
Final recommendation: Reconsider after a major revision.
Comments on the Quality of English Language
Dear Authors,
Reviewer comments antioxidants-2819707
The manuscript entitled „Antioxidant agriculture for stress-resilient crop production: The field practice“ represents a useful review manuscript on the concept of so-called antioxidant agriculture as a whole of means and practices aimed to alleviate the adverse effects of oxidative stress on crop plants. The review manuscript provides data on the utilization of genetic engineering, application of several bioactive compounds including antioxidants, phytohormones, plant biostimulants, and agronomic practice to improve crop tolerance to oxidative stress. A comprehensive overview of the studies using various antioxidant agents to alleviate adverse impacts of environmental stresses, especially drought, high salinity, and low temperature in a wide range of crops including alfalfa, barley, wheat, rice, maize, common bean, sugar beet, and rapeseeds is provided in Table 1.
I think that the manuscript summarises very useful data importnat for further research and thus it deserves to be published in Antioxidants.
However, I have a few important comments on the present version of the manuscript which are provided below:
The concept of antioxidant argiculture discussed in the text should be described more clearly. What are the inputs and the outputs of antioxidant agriculture? In addition to the text and Table 1 providing just an overview on published literature on this topic, I would strongly recommend the authors to add a figure (Figure 1) providing a scheme summarising the concept of antioxidant agriculture considering its inputs (genetic engineering, treatments with antioxidant agents and bioactive compounds, agricultural management practices) and outputs (improved crop stress tolerance).
Formal comments on the text related to terminology, English language and style:
Lines 154-155: Use „gamma“ letter instead of „g“ for „γ-glutamylcysteine synthetase (γ-GCS)“, not „g-ECS“ – please, correct the spelling of the abbreviation commonly used for the enzyme!
Line 192: Add either the word „fact“ or „finding“ in the statement: „This finding suggests that conventional breeding programs can play a significant role in….“
Line 258: Add the word „finding“ following „This“ in the statement: „This finding raises concerns about potential cost limitations…“
Line 278: Briefly define „white sugar content“, i.e., „sucrose“???
Line 297: The sceintific name for „seaweeds“ has to be given.
Line 306: Add the word „process“ in the statement „This proces resulted in a 3.08 t/ha increase….“
Line 315: The term „leonardite“ (containing up to 90% humic acid) probably refers to some commercial means used in the study. Therefore, an appropriate reference including the manufacturer – company name – has to be given for leonardite.
Line 389: Add the word „aim“ following „this“ in the statement „To acheive this aim,….“
Final recommendation: Reconsider after a major revision.
Reviewer 2 Report
Comments and Suggestions for Authors
This opinion manuscript deals with a new concept called "antioxidant agriculture". It encompasses the most recent references with field experiments to study and alleviate ROS production by different ways.
This idea is interesting and the approach is achieved. The manuscript is easy to read. It fully meets Antioxidants expectations.
This manuscript contains three debatable points
1- the concept's name Antioxidants agriculture. In fact, it can be confused with antioxidant crops, which are well known by now. I suggest changing it to antioxidant agricultural management.
2-For better reading and comprehension, I propose to extend figure 1 by including the enzymes and their sites of action.
3- what about use of essential oil as biostimulant or as curative means to overcome stresses?
Comments on the Quality of English LanguageThe manuscript is really eay to read. some minor typo modifications are necessary
Reviewer 3 Report
Comments and Suggestions for Authors
The ms is average written but it is more suitable as a short communication and not an opinion article. The target of the ms to propose antioxidant agriculture is not only a good presentation of the literature but requires in-depth discussion and proposal of important pillars to achieve the targets of antioxidant agriculture.
Comments on the Quality of English LanguageMinor corrections are required
Round 2
Reviewer 1 Report
Comments and Suggestions for Authors
Dear Authors,
Reviewer comments antioxidants-2819707.R1
The quality of the revised manuscript entitled „Antioxidant agriculture for stress-resislient crop production: The field practice“ was improved by the authors in accordance with my previous comments. A definition of the concept of antioxidant agriculture and a comprehensive scheme on the topic of the opinion article were added to the manuscript as a relevant text and Figure 1, respectively.
I have only a few formal comments on the revised manuscript which are provided below:
Line 61: Modify the word form „removing“ to „removal“ in the statement: „Therefore, plants must maintain a delicate balance between ROS production and removal in order to avoid oxidative stress.“
Line 226: Modify the word form „vegetable“ to „vegetative“ in the term „vegetative plant growth“ in the statement: „…significantly improved vegetative plant growth and green pods yield in drought-stressed fields.“
Line 245: Add „a“ preceding the word „decrease“ in the statement „…leading to a decrease in ROS and MDA levels.“
Line 255: Use rather the name „common bean“ than „snap bean“ for „Phasolus vulgaris“ in a scientific text since „snap bean“ is more associated with culinary conotations.
Final recommendation: Accept after a minor revision.
Comments on the Quality of English Language
Dear Authors,
Reviewer comments antioxidants-2819707.R1
The quality of the revised manuscript entitled „Antioxidant agriculture for stress-resislient crop production: The field practice“ was improved by the authors in accordance with my previous comments. A definition of the concept of antioxidant agriculture and a comprehensive scheme on the topic of the opinion article were added to the manuscript as a relevant text and Figure 1, respectively.
I have only a few formal comments on the revised manuscript which are provided below:
Line 61: Modify the word form „removing“ to „removal“ in the statement: „Therefore, plants must maintain a delicate balance between ROS production and removal in order to avoid oxidative stress.“
Line 226: Modify the word form „vegetable“ to „vegetative“ in the term „vegetative plant growth“ in the statement: „…significantly improved vegetative plant growth and green pods yield in drought-stressed fields.“
Line 245: Add „a“ preceding the word „decrease“ in the statement „…leading to a decrease in ROS and MDA levels.“
Line 255: Use rather the name „common bean“ than „snap bean“ for „Phasolus vulgaris“ in a scientific text since „snap bean“ is more associated with culinary conotations.
Final recommendation: Accept after a minor revision.
Author Response
We sincerely appreciate the reviewer very much for his/her kind and detailed comments. We have addressed all the points raised with gratitude. Thank you!
Reviewer 2 Report
Comments and Suggestions for Authors
Dear Authors,
Thank you for condidering positively the remaks done on your manuscript. Thank you for your modifications. Some of the explanations concerning antioxidant agriculture, while debatable, are still acceptable.
Author Response
We thank the reviewer very much for the positive and encouraging comments. Thank you!